# Hierarchical Representation Matching for CLIP-based Class-Incremental Learning

## Abstract

Class-Incremental Learning (CIL) aims to endow models with the ability to continuously adapt to evolving data streams. Recent advances in pre-trained vision-language models (e.g., CLIP) provide a powerful foundation for this task. However, existing approaches often rely on simplistic templates, such as "a photo of a [CLASS]", which overlook the hierarchical nature of visual concepts. For example, recognizing *"cat"* versus *"car"* depends on coarse-grained cues, while distinguishing *"cat"* from *"lion"* requires fine-grained details. Similarly, the current feature mapping in CLIP relies solely on the representation from the last layer, neglecting the hierarchical information contained in earlier layers. In this work, we introduce HiErarchical Representation MAtchiNg (HERMAN) for CLIP-based CIL. Our approach leverages LLMs to recursively generate discriminative textual descriptors, thereby augmenting the semantic space with explicit hierarchical cues. These descriptors are matched to different levels of the semantic hierarchy and adaptively routed based on task-specific requirements, enabling precise discrimination while alleviating catastrophic forgetting in incremental tasks. Extensive experiments on multiple benchmarks demonstrate that our method consistently achieves state-of-the-art performance.

## 1 Introduction

Deep learning (Hinton et al., 2006; Deng et al., 2009; LeCun et al., 2015; He et al., 2015; Goodfellow et al., 2016) has achieved remarkable progress across domains. However, real-world data streams pose unique challenges, particularly due to the need for continuous adaptation to new information. Class-Incremental Learning (CIL) (Zhao et al., 2020; Masana et al., 2022; Yu et al., 2024a; Dohare et al., 2024; Wang et al., 2024; Lai et al., 2025) addresses this by gradually incorporating new categories, but a major obstacle is catastrophic forgetting (Serra et al., 2018; Ramasesh et al., 2021; Shi et al., 2021), where learning new classes diminishes previously acquired knowledge of old classes. Early approaches often train models from scratch (Li & Hoiem, 2018; Rebuffi et al., 2017b; Yan et al., 2021), whereas recent work leverages pre-trained models (PTMs) (Wang et al., 2022c;b; Smith et al., 2023; Huang et al., 2024) such as CLIP (Radford et al., 2021). By leveraging rich prior knowledge from large-scale training, PTMs provide stronger initialization and support more effective continual adaptation, making them well-suited for real-world incremental learning.

As a representative PTM, CLIP (Radford et al., 2021) aligns images with textual descriptors and has demonstrated impressive zero-shot generalization. However, the commonly used fixed templates such as "a photo of a [CLASS]" capture only coarse semantics and fail to encode richer hierarchical textual information (Menon & Vondrick, 2023; Khattak et al., 2023; Liu et al., 2024). Coarse-grained descriptors, such as *"animal"* versus *"vehicle"*, are sufficient to distinguish semantically distant categories like *"cat"* and *"car"*. In contrast, separating semantically similar categories such as *"cat"* and *"lion"* requires fine-grained descriptors, such as *"soft and finely textured fur"* and *"thick and elongated whiskers"*. Relying solely on class names neglects these discriminative cues, which in turn limits cross-modal alignment. Even when learnable prompt tuning is employed, prompts are still modeled as single-level representations without explicit control over semantic hierarchy.

Similarly, the hierarchical structure of visual representations in CLIP is often overlooked (Maniparambil et al., 2023; Novack et al., 2023; Pratt et al., 2023). Most existing feature mapping strategies rely exclusively on the representation from the last layer, which emphasizes global semantics

but ignores the progressively refined information encoded in intermediate layers. These intermediate features naturally capture multiple levels of abstraction, from low-level textures to high-level object semantics, and thus provide complementary signals that are crucial for robust CIL.

However, even with hierarchical textual and visual representations, a key challenge remains: different hierarchies contribute unequally during inference. Coarse-grained semantics, such as *"animal"*, help with discriminating semantically distant categories. Fine-grained cues, like *"thick and elongated whiskers"*, are crucial for distinguishing closely related categories. Uniformly treating all levels risks impairing critical details or overemphasizing irrelevant cues (Wu et al., 2024a; Dai et al., 2024; Lin et al., 2025). To address this, it is crucial to design a mechanism that can dynamically exploit hierarchical representations (Tu et al., 2023). This enables more accurate visual-textual alignment. Moreover, as tasks arrive continually, an additional strategy is required to facilitate the acquisition of new knowledge, stabilize the routing process, and alleviate catastrophic forgetting.

Learning with hierarchical representations presents a promising avenue for strengthening CLIP in CIL. Nevertheless, two key challenges remain: **1)** how to obtain and organize hierarchical representations. Such hierarchical semantics are essential because they capture both coarse-grained cues and fine-grained attributes, thereby improving recognition accuracy and reducing the risk of forgetting; **2)** how to dynamically route and balance the contributions of hierarchical representations. Fine-grained routing control allows the model to emphasize the most informative cues for each task, improving adaptability while mitigating catastrophic forgetting in CIL.

To tackle these challenges, we propose HiErarchical Representation MAtchiNg (HERMAN) for CLIP-based CIL. Our framework first leverages LLMs (OpenAI, 2025; Yang et al., 2025) to recursively generate discriminative textual descriptors at multiple hierarchical levels and explicitly match them to corresponding visual features. This alignment constructs a structured semantic space where both coarse- and fine-grained cues are faithfully associated with their visual counterparts. In addition to this enriched space, we introduce an adaptive routing mechanism that dynamically allocates weights across the semantic hierarchy, enabling the model to emphasize the most relevant features for each input. To further stabilize routing and avoid catastrophic forgetting, the router is updated through a projection that preserves informative subspaces of prior knowledge while allowing compact adaptation to new categories. By jointly enriching, matching, and adaptively routing hierarchical semantics, HERMAN enhances the discriminative power of CLIP in CIL, mitigates forgetting, and achieves consistent gains across standard benchmarks.

## 2 RELATED WORK

**Pre-Trained Model-Based CIL.** Early approaches to class-incremental learning (CIL) (Rebuffi et al., 2017b; Li & Hoiem, 2018; De Lange et al., 2021; Yan et al., 2021; Masana et al., 2022) typically trained models from scratch, which often led to limited generalization and severe forgetting when adapting to new tasks. To overcome these drawbacks, recent studies have shifted towards leveraging large-scale pre-trained models (PTMs), whose rich prior knowledge provides a stronger foundation for continual learning (Wang & Huang, 2024; Seo et al., 2024; Yu et al., 2025; van de Ven, 2025). A common strategy is to freeze the pre-trained backbone and introduce lightweight modules, such as prompts (Wang et al., 2022b;c;a; Zhou et al., 2022; Smith et al., 2023) and adapters (Rebuffi et al., 2017a; Chen et al., 2022; Yu et al., 2024a). For example, L2P (Wang et al., 2022c) and DualPrompt (Wang et al., 2022b) maintain a pool of visual prompts (Jia et al., 2022) and dynamically select instance-specific prompts for each input when fine-tuning a pre-trained Vision Transformer. Other methods design more sophisticated prompt strategies, such as combining or generating prompts through attention mechanisms (Smith et al., 2023; Wang et al., 2023a; Li & Zhou, 2025) or generative networks (Jung et al., 2023). Prototype-based methods further exploit PTM features by directly constructing classifiers via class prototypes (Snell et al., 2017; Zhou et al., 2025a; McDonnell et al., 2023). When CLIP is adopted as initialization, multi-modal prompts are explored to better align vision and language representations (Wang et al., 2022a; Liang et al., 2022; Wang et al., 2023b). Beyond prompts, MOE-Adapter (Yu et al., 2024a) extends lightweight module selection with a mixture-of-experts design (Masoudnia & Ebrahimpour, 2014), while RAPF (Huang et al., 2024) enhances adapters through decomposed parameter fusion to mitigate forgetting.

**Textual descriptors for CLIP.** PTMs, such as CLIP (Radford et al., 2021), align images and texts in a shared embedding space, achieving strong zero-shot recognition and retrieval. However, their performance in specialized domains, such as healthcare, remains limited due to the need for fine-grained visual cues and domain-specific semantics. To address this, a variety of lightweight adaptation strategies have been proposed, with text prompting attracting particular attention. Existing approaches can be broadly categorized into two main types: soft prompting and hard prompting. Soft prompting (Zhou et al., 2022; Wu et al., 2024b; Fu et al., 2024; Tian et al., 2024; Zhang et al., 2024) introduces learnable tokens into the textual input, enabling parameter-efficient adaptation while keeping the pre-trained backbone frozen. In contrast, hard prompting (Wen et al., 2023; Yu et al., 2024b) directly inserts natural language phrases into predefined templates, requiring no additional training and offering better interpretability. However, both soft and hard prompting still fall short in capturing the hierarchical representations that are crucial for CIL. Motivated by this limitation, we propose to extend hard prompting by introducing hierarchical textual descriptors. These descriptors serve as semantically rich anchors that transcend conventional flat templates, allowing models to more effectively capture and enhance recognition in hierarchical representation spaces.

## 3 PRELIMINARIES

### 3.1 PROBLEM SETUP: CLASS-INCREMENTAL LEARNING

CIL aims to continually extend a unified classifier as novel classes arrive in a sequence of tasks (Rebuffi et al., 2017b). Let the training stream be $\{D_1, D_2, \ldots, D_T\}$, where each incremental task $D_t = \{(\mathbf{x}_i, y_i)\}_{i=1}^{n_t}$ contains $n_t$ labeled instances. Each input $\mathbf{x}_i \in \mathcal{X}$ belongs to a class $y_i \in Y_t$, where $Y_t$ is the label set of task $t$. We assume disjoint label spaces across tasks, *i.e.*, $Y_t \cap Y_{t'} = \varnothing$ for $t \neq t'$. Following the *exemplar-free* protocol (Zhu et al., 2021; Wang et al., 2022b;c), the learner cannot store or replay samples from past tasks. When learning the $t$-th task, only $D_t$ is accessible. The objective is then to build a unified classifier over all classes observed so far, $\mathcal{Y}_t = Y_1 \cup \cdots \cup Y_t$, by finding a function $f : \mathcal{X} \to \mathcal{Y}_t$ that minimizes the empirical risk:

$$f^\star = \arg\min_{f \in \mathcal{H}} \, \mathbb{E}_{(x,y) \sim D_1 \cup \cdots \cup D_t} \, \mathbb{I}\big(y \neq f(x)\big), \tag{1}$$

where $\mathcal{H}$ denotes the hypothesis space and $\mathbb{I}(\cdot)$ is the indicator function.

### 3.2 CLIP-BASED CLASSIFICATION

Following Zhou et al. (2025b), we assume a pre-trained CLIP is available for classification. Given an input image $\mathbf{x} \in \mathbb{R}^{H \times W \times C}$, the vision encoder $g_v(\cdot)$ first partitions it into a sequence of flattened 2D patches $\mathbf{x}_e \in \mathbb{R}^{(L-1) \times D}$, where $(L-1)$ denotes the number of patch tokens and $D$ is the embedding dimension. A learnable [CLS] token $\mathbf{x}_{\text{cls}} \in \mathbb{R}^D$ is prepended to obtain the sequence $\mathbf{x}_p = [\mathbf{x}_{\text{cls}}; \mathbf{x}_e]$, which is processed by $B$ layers. The representation at the $b$-th layer is:

$$\mathbf{x}^b = [\mathbf{x}_{\text{cls}}^b; \mathbf{x}_1^b; \ldots; \mathbf{x}_{L-1}^b] \in \mathbb{R}^{L \times D}, \tag{2}$$

where $\mathbf{x}_{\text{cls}}^b \in \mathbb{R}^D$ is the class token and $\mathbf{x}_l^b$ is the patch token.

For each class $i \in \mathcal{Y}_t$, we utilize a templated prompt $\mathbf{c}_i$ (*e.g.*, "a photo of a [CLASS]") and obtain its textual embedding $\mathbf{e}_i = g_t(\mathbf{c}_i)$ via the text encoder (Radford et al., 2021). Classification is performed by measuring the similarity between the final-layer [CLS] token $\mathbf{x}_{\text{cls}}^B$ and these textual embeddings. The probability of class $i$ is then obtained as a Softmax over the cosine similarities:

$$f_{y_i}(\mathbf{x}) = \frac{\exp\big(\cos(\mathbf{x}_{\text{cls}}^B, \mathbf{e}_i)\big)}{\sum_{j=1}^{|\mathcal{Y}_t|} \exp\big(\cos(\mathbf{x}_{\text{cls}}^B, \mathbf{e}_j)\big)} = \frac{\exp\big(\cos(g_v(\mathbf{x}), g_t(\mathbf{c}_i))\big)}{\sum_{j=1}^{|\mathcal{Y}_t|} \exp\big(\cos(g_v(\mathbf{x}), g_t(\mathbf{c}_j))\big)}, \tag{3}$$

where $\cos(\cdot, \cdot)$ denotes cosine similarity. In this formulation, CLIP performs classification by aligning visual embeddings with textual prompts, which can be seamlessly extended to CIL by enlarging the label space $\mathcal{Y}_t$ as new categories arrive (Huang et al., 2024; Yu et al., 2024a).

**Discussions.** As shown in Eq. 3, classification depends only on the final-layer visual embedding aligned with a simple templated prompt. However, the hierarchical representations generated by the

vision encoder $g_v(\cdot)$ in Eq. 2 are overlooked. These intermediate embeddings encode hierarchical semantics that can further enhance the final representation. Likewise, textual representations should not be constrained to fixed templates; instead, they should also exhibit hierarchical structures. By incorporating hierarchical visual and textual representations, it becomes possible to perform cross-modal alignment in a hierarchical view. Furthermore, an effective mechanism is required to dynamically exploit such hierarchical information while continually updating representations without incurring catastrophic forgetting.

## 4 HERMAN: HIERARCHICAL REPRESENTATION MATCHING

To address the observed challenges, we introduce hierarchical textual descriptors that mirror the multi-level structure encoded in intermediate visual representations. These descriptors are aligned with visual features at corresponding hierarchical levels, providing auxiliary supervision throughout the hierarchy and strengthening vision–language alignment. In addition to this enriched space, we further employ an adaptive routing mechanism that regulates the contributions of different hierarchical levels for each input. To stabilize routing and alleviate catastrophic forgetting, the router is updated through a projection that preserves informative subspaces of prior knowledge while enabling compact adaptation to new categories.

### 4.1 OBTAINING HIERARCHICAL REPRESENTATIONS

In visual recognition, the semantic information required for discrimination is not uniform across categories. Coarse-grained descriptors, such as *"animal"* versus *"vehicle"*, are sufficient to separate semantically distant classes like *"cat"* and *"car"*. In contrast, distinguishing semantically similar categories such as *"cat"* and *"lion"* requires fine-grained descriptors, such as *"soft and finely textured fur"* or *"thick and elongated whiskers"*. Conventional CLIP, however, relies on a single fixed prompt aligned with the final-layer representation, overlooking such hierarchical cues. To address this, we introduce hierarchical textual descriptors that range from coarse to fine-grained levels, aligning them with intermediate visual features to provide richer supervision.

To capture semantic information at different levels of abstraction, we construct a set of natural language descriptors for each class that spans from coarse-grained cues to fine-grained details. Concretely, for each class $i \in \mathcal{Y}_t$, we *recursively* generate a collection of $M$ descriptors $\mathcal{T}_i = \{\mathbf{t}_{i,1}, \ldots, \mathbf{t}_{i,M}\}$ with the aid of LLMs. These descriptions are designed to progress from high-level category cues to increasingly detailed characteristics. To illustrate, we take the class *"Aston_Martin_Virage_Convertible_2012"* from StanfordCars (Krause et al., 2013) as an example:

**Q:** *What hierarchical visual features characterize [CLASS]$_i$ in an image?*

**A:** *There are several useful visual features to tell there is a [CLASS]$_i$ in an image:*

- *sports car*
- *sculpted side panels with recessed door handles*
- *contrasting textures of glossy paint, metallic rims, and matte grille details*

At the coarsest level, the descriptors emphasize only the overall identity of the object, while intermediate ones highlight representative architectural elements, and the finest-grained descriptors capture rich contextual details by combining multiple attributes together. Once generated, these descriptors are mapped into a shared embedding space through the CLIP text encoder $g_t(\cdot)$:

$$\mathbf{z}_{i,m} = g_t(\mathbf{t}_{i,m}) \in \mathbb{R}^D, \quad m = 1, \ldots, M, \tag{4}$$

which yields a set of textual embeddings that capture different levels of semantic abstraction, forming a hierarchical textual space that grounds the model for subsequent alignment with visual features.

To enable such alignment, the vision encoder $g_v(\cdot)$ processes an input image $\mathbf{x}$ through $B$ layers, producing intermediate representations as defined in Eq. 2. At each layer $b$, we extract the [CLS] token $\mathbf{x}_{\text{cls}}^b \in \mathbb{R}^D$, which serves as the summary of visual information at the corresponding hierarchical level of representation, and match it with the textual embedding $\mathbf{z}_{i,m}$ associated with class $i$. In other words, the collection of hierarchical visual embeddings $\{\mathbf{x}_{\text{cls}}^b\}_{b=1}^B$ is jointly aligned with the

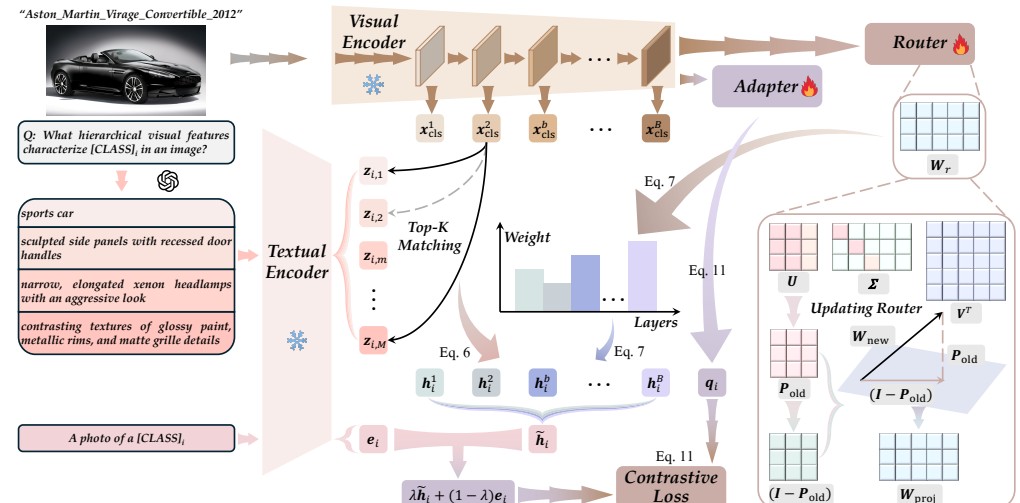

Figure 1: Illustration of HERMAN. The model leverages LLMs to generate hierarchical textual descriptors and explicitly matches them with intermediate visual features of CLIP. An adaptive routing mechanism then balances contributions across different hierarchical levels, while a projection-based update stabilizes routing to alleviate catastrophic forgetting in continual learning.

set of hierarchical textual descriptors $\{\mathbf{z}_{i,m}\}_{m=1}^{M}$, establishing cross-modal alignment across multiple hierarchical representations. We therefore compute the cosine similarity $s_{i,m}^{b}$ between $\mathbf{x}_{\text{cls}}^{b}$ and $\mathbf{z}_{i,m}$, which provides a quantitative measure of their alignment in the shared hierarchy space:

$$s_{i,m}^{b} = \cos\left(\mathbf{x}_{\text{cls}}^{b}, \mathbf{z}_{i,m}\right). \tag{5}$$

However, simply using all descriptors directly may introduce redundancy and noise. To mitigate this, we retain only the Top-$K$ descriptors and normalize their weights. The aligned descriptors are then aggregated to form the united hierarchical textual embeddings for class $i$ at layer $b$:

$$\mathbf{h}_{i}^{b} = \sum_{k} \alpha_{i,k}^{b} \, \mathbf{z}_{i,k}, \quad \text{where } \alpha_{i,k}^{b} = \text{Softmax}\left(\text{Top}_{K}\left(\{s_{i,k}^{b}\}_{k=1}^{K}\right)\right). \tag{6}$$

In Eq. 6, $\mathbf{h}_{i}^{b}$ represents a compact textual representation corresponding to $\mathbf{x}_{\text{cls}}^{b}$. By grounding intermediate visual features in their most relevant textual descriptors, the model benefits from hierarchical supervision that extends beyond the final-layer alignment in Eq. 3. As illustrated in Fig. 1, this design enforces cross-modal alignment of hierarchical visual-textual representations, thereby preserving semantic coherence across levels, from coarse-grained cues to fine-grained details.

## 4.2 ADAPTIVE HIERARCHY ROUTING

While such hierarchical alignment establishes meaningful correspondences between intermediate visual and textual representations, an open question remains: how should these hierarchical representations be integrated to form the final prediction? A naïve strategy would be to average the representations across different layers, but this overlooks the fact that the relative importance of hierarchical levels can vary across tasks and inputs, with certain representations being more decisive for discrimination than others. To overcome this limitation, we introduce an adaptive routing mechanism that dynamically regulates the contributions of different hierarchical levels. Crucially, this routing must remain flexible as new tasks arrive, while also preserving knowledge from previously learned tasks to mitigate catastrophic forgetting.

Specifically, to adaptively regulate the relative contributions of hierarchical representations, we employ a router implemented as a lightweight linear layer $\mathbf{W}_{r} \in \mathbb{R}^{B \times D}$. Conditioned on the final-layer visual representation $\mathbf{x}_{\text{cls}}^{B}$, the router dynamically generates a set of non-negative weights $\boldsymbol{\beta}_{i} = \{\beta_{i}^{1}, \ldots, \beta_{i}^{B}\}$ for the $B$ unified hierarchical textual embeddings of class $i$, and the final representation is subsequently obtained as their convex combination:

$$\tilde{\mathbf{h}}_{i} = \sum_{b=1}^{B} \beta_{i}^{b} \, \mathbf{h}_{i}^{b}, \quad \text{where } \boldsymbol{\beta}_{i} = \text{Softmax}\left(\mathbf{W}_{r}\mathbf{x}_{\text{cls}}^{B}\right). \tag{7}$$

Therefore, the model can adaptively exploit different hierarchies depending on the input, thereby enhancing discrimination. Nonetheless, continually updating the router may overwrite routing patterns established for previous tasks, leading to catastrophic forgetting. To mitigate this, we update the router under a projection constraint that retains knowledge-rich subspaces, thus stabilizing learning across tasks. Specifically, after each incremental task, the weight of $\mathbf{W}_r$ is decomposed as:

$$\mathbf{W}_r = \mathbf{U}\boldsymbol{\Sigma}\mathbf{V}^\top, \tag{8}$$

where $\mathbf{U} \in \mathbb{R}^{B \times B}$ and $\mathbf{V} \in \mathbb{R}^{D \times D}$ contain the left and right singular vectors, and $\boldsymbol{\Sigma} = \mathrm{diag}(\sigma_1, \ldots, \sigma_B)$ with $\sigma_b \geq 0$. We select the smallest rank $B'$ that preserves at least a proportion $\delta$ of the cumulative energy and form the projection matrix onto the retained subspace:

$$\mathbf{P}_{\mathrm{old}} = \mathbf{U}_{[:,1:B']}\mathbf{U}_{[:,1:B']}^\top \in \mathbb{R}^{B \times B}, \quad \text{where } B' = \min\left\{ r \ \middle| \ \frac{\sum_{b=1}^{r} \sigma_b^2}{\sum_{b=1}^{B} \sigma_b^2} \geq \delta \right\}, \ \delta \in (0, 1]. \tag{9}$$

where $\mathbf{P}_{\mathrm{old}}$ serves as an orthogonal projector encoding the routing pattern subspace accumulated for old tasks. During parameter updates, let $\mathbf{W}_{\mathrm{new}} \in \mathbb{R}^{B \times D}$ denote the router weight obtained by optimizing on the current task. We project $\mathbf{W}_{\mathrm{new}}$ into components within and outside the preserved subspace and compute the projected update with $\mathbf{P}_{\mathrm{old}}$:

$$\mathbf{W}_{\mathrm{proj}} = \rho\, \mathbf{P}_{\mathrm{old}}\, \mathbf{W}_{\mathrm{new}} + (1 - \rho)\, (\mathbf{I} - \mathbf{P}_{\mathrm{old}})\, \mathbf{W}_{\mathrm{new}}, \tag{10}$$

where $\mathbf{I} \in \mathbb{R}^{B \times B}$ is the identity matrix and $\rho \in [0, 1]$ balances stability and plasticity. In Eq. 10, the first term preserves and reinforces directions already captured by previous tasks, as it retains the component of $\mathbf{W}_{\mathrm{new}}$ lying in the preserved subspace, whereas the second term instead extracts the component in the orthogonal complement, allowing the router to capture novel information and adapt to new tasks without disrupting prior knowledge. We then initialize the next task with $\mathbf{W}_{\mathrm{proj}}$, which stabilizes routing dynamics across tasks and enables the model to flexibly integrate new knowledge while maintaining long-term consistency in continual learning.

### 4.3 Cross-Modal Matching

While the router integrates hierarchical textual representations, it remains essential to establish a joint space where visual and textual features can be directly compared and contrasted. To this end, we introduce an adapter $\mathbf{W}_a \in \mathbb{R}^{D \times D}$ that projects $\mathbf{x}_{\mathrm{cls}}^B$ into the joint space, yielding $\mathbf{q}_i = \mathbf{W}_a \mathbf{x}_{\mathrm{cls}}^B$ for cross-modal alignment. To enhance textual supervision, we mix the aggregated textual embedding $\tilde{\mathbf{h}}_i$ with the templated embedding $\mathbf{e}_i = g_t(\mathbf{t}_i)$ through a convex interpolation controlled by $\lambda \in [0, 1]$. Building on this, we reformulate the prediction of class $i$ in Eq. 3 using the adapted visual embedding and the mixed textual embedding, which yields the following contrastive loss:

$$\mathcal{L} = \ell(f_{y_i}(\mathbf{x}), y_i), \quad \text{where } f_{y_i}(\mathbf{x}) = \frac{\exp\big(\cos(\mathbf{q}_i, (\lambda\tilde{\mathbf{h}}_i + (1-\lambda)\mathbf{e}_i)/\tau)\big)}{\sum_{j=1}^{|\mathcal{Y}_t|} \exp\big(\cos(\mathbf{q}_i, (\lambda\tilde{\mathbf{h}}_j + (1-\lambda)\mathbf{e}_j)/\tau)\big)}. \tag{11}$$

To further mitigate forgetting, we adopt feature-level generative replay: rather than reconstructing raw images, we model the distribution of $\mathbf{x}_{\mathrm{cls}}^B$ for each class $i$ as a Gaussian $\mathcal{N}(\boldsymbol{\mu}_i, \boldsymbol{\Sigma}_i)$ estimated from the empirical mean and covariance (Zhu et al., 2021). For intermediate layers, only the diagonal covariance is retained to reduce memory cost. During incremental training, pseudo-features sampled from these Gaussians act as compact surrogates for past classes, alleviating catastrophic forgetting.

**Summary of** HERMAN**.** We enhance CLIP-based CIL by introducing LLM-generated hierarchical textual descriptors, which are encoded into embeddings (Eq. 4), aligned with intermediate visual features (Eq. 5), and further aggregated into unified hierarchical representations (Eq. 6). A router convexly combines these embeddings (Eq. 7) and is updated with a projection constraint to effectively mitigate forgetting (Eq. 10). During training, the model is jointly optimized with Eq. 11. During inference, final predictions are obtained by matching visual embeddings against the mixed textual embeddings of all previously seen classes using the same formulation.

## 5 Experiments

In this section, we evaluate HERMAN on nine benchmark datasets and compare it with state-of-the-art approaches. In addition, we conduct ablation studies and parameter sensitivity analysis, and provide visualizations to further validate the robustness and interpretability of our framework.

Table 1: Average and last performance comparison of different methods. The best performance is shown in bold. All methods are initialized with the same pre-trained CLIP for a fair comparison.

| Method | Aircraft | | | | Cars | | | | CIFAR100 | | | |
| | B0 Inc10 | | B50 Inc10 | | B0 Inc10 | | B50 Inc10 | | B0 Inc10 | | B50 Inc10 | |
| | $\bar{\mathcal{A}}$ | $\mathcal{A}_B$ | $\bar{\mathcal{A}}$ | $\mathcal{A}_B$ | $\bar{\mathcal{A}}$ | $\mathcal{A}_B$ | $\bar{\mathcal{A}}$ | $\mathcal{A}_B$ | $\bar{\mathcal{A}}$ | $\mathcal{A}_B$ | $\bar{\mathcal{A}}$ | $\mathcal{A}_B$ |
|---|---|---|---|---|---|---|---|---|---|---|---|---|
| Finetune | 3.16 | 0.96 | 1.72 | 1.05 | 3.14 | 1.10 | 1.54 | 1.13 | 7.84 | 4.44 | 5.30 | 2.46 |
| SimpleCIL (Zhou et al., 2025a) | 59.24 | 48.09 | 53.05 | 48.09 | 92.04 | 86.85 | 88.96 | 86.85 | 84.15 | 76.63 | 80.20 | 76.63 |
| CoOp (Zhou et al., 2022) | 14.54 | 7.14 | 13.05 | 7.77 | 36.46 | 21.65 | 37.40 | 20.87 | 47.00 | 24.24 | 41.23 | 24.12 |
| ZS-CLIP (Radford et al., 2021) | 26.66 | 17.22 | 21.70 | 17.22 | 82.60 | 76.37 | 78.32 | 76.37 | 81.81 | 71.38 | 76.49 | 71.38 |
| L2P (Wang et al., 2022c) | 47.19 | 28.29 | 44.07 | 32.13 | 76.63 | 61.82 | 76.37 | 65.64 | 82.74 | 73.03 | 81.14 | 73.61 |
| DualPrompt (Wang et al., 2022b) | 44.30 | 25.83 | 46.07 | 33.57 | 76.26 | 62.94 | 76.88 | 67.55 | 81.63 | 72.44 | 80.12 | 72.57 |
| CODA-Prompt (Smith et al., 2023) | 45.98 | 27.69 | 45.14 | 32.28 | 80.21 | 66.47 | 75.06 | 64.19 | 82.43 | 73.43 | 78.69 | 71.58 |
| RAPF (Huang et al., 2024) | 50.38 | 23.61 | 40.47 | 25.44 | 82.79 | 71.22 | 77.21 | 69.97 | 86.14 | 78.04 | 82.17 | 77.93 |
| PROOF (Zhou et al., 2025b) | 63.81 | 56.14 | 59.47 | 57.10 | 90.74 | 86.51 | 88.00 | 85.58 | 86.77 | 79.11 | 83.32 | 79.73 |
| HERMAN | **66.70** | **58.84** | **60.41** | **58.18** | **92.49** | **89.13** | **89.68** | **89.00** | **89.02** | **83.21** | **85.91** | **82.98** |

| Method | Food | | | | UCF | | | | CUB | | | |
| | B0 Inc10 | | B50 Inc10 | | B0 Inc10 | | B50 Inc10 | | B0 Inc20 | | B100 Inc20 | |
| | $\bar{\mathcal{A}}$ | $\mathcal{A}_B$ | $\bar{\mathcal{A}}$ | $\mathcal{A}_B$ | $\bar{\mathcal{A}}$ | $\mathcal{A}_B$ | $\bar{\mathcal{A}}$ | $\mathcal{A}_B$ | $\bar{\mathcal{A}}$ | $\mathcal{A}_B$ | $\bar{\mathcal{A}}$ | $\mathcal{A}_B$ |
|---|---|---|---|---|---|---|---|---|---|---|---|---|
| Finetune | 3.49 | 1.71 | 2.14 | 1.52 | 4.51 | 1.59 | 1.21 | 0.80 | 2.06 | 0.64 | 0.56 | 0.47 |
| SimpleCIL (Zhou et al., 2025a) | 87.89 | 81.65 | 84.73 | 81.65 | 90.44 | 85.68 | 88.12 | 85.68 | 83.81 | 77.52 | 79.75 | 77.52 |
| CoOp (Zhou et al., 2022) | 36.01 | 14.18 | 33.13 | 18.67 | 47.85 | 33.46 | 42.02 | 24.74 | 27.61 | 8.57 | 24.03 | 10.14 |
| ZS-CLIP (Radford et al., 2021) | 87.86 | 81.92 | 84.75 | 81.92 | 75.50 | 67.64 | 71.44 | 67.64 | 74.38 | 63.06 | 67.96 | 63.06 |
| L2P (Wang et al., 2022c) | 85.66 | 77.33 | 80.42 | 73.13 | 86.34 | 76.43 | 83.95 | 76.62 | 70.87 | 57.93 | 75.64 | 66.12 |
| DualPrompt (Wang et al., 2022b) | 84.92 | 77.29 | 80.00 | 72.75 | 85.21 | 75.82 | 84.31 | 76.35 | 69.89 | 57.46 | 74.40 | 64.84 |
| CODA-Prompt (Smith et al., 2023) | 86.18 | 78.78 | 80.98 | 74.13 | 87.76 | 80.14 | 83.04 | 75.03 | 73.12 | 62.98 | 73.95 | 62.21 |
| RAPF (Huang et al., 2024) | 88.57 | 81.15 | 85.53 | 81.17 | 92.28 | 80.33 | 90.31 | 81.55 | 79.09 | 62.77 | 72.82 | 62.93 |
| PROOF (Zhou et al., 2025b) | 90.04 | 84.73 | 87.52 | 84.74 | 94.58 | 91.10 | 93.58 | 90.91 | 82.31 | 76.64 | 79.20 | 76.37 |
| HERMAN | **90.93** | **86.18** | **88.53** | **86.26** | **97.69** | **95.72** | **96.63** | **95.83** | **84.71** | **78.37** | **80.52** | **77.95** |

| Method | ImageNet-R | | | | ObjectNet | | | | SUN | | | |
| | B0 Inc20 | | B100 Inc20 | | B0 Inc20 | | B100 Inc20 | | B0 Inc30 | | B150 Inc30 | |
| | $\bar{\mathcal{A}}$ | $\mathcal{A}_B$ | $\bar{\mathcal{A}}$ | $\mathcal{A}_B$ | $\bar{\mathcal{A}}$ | $\mathcal{A}_B$ | $\bar{\mathcal{A}}$ | $\mathcal{A}_B$ | $\bar{\mathcal{A}}$ | $\mathcal{A}_B$ | $\bar{\mathcal{A}}$ | $\mathcal{A}_B$ |
|---|---|---|---|---|---|---|---|---|---|---|---|---|
| Finetune | 1.37 | 0.43 | 1.01 | 0.88 | 1.34 | 0.47 | 0.69 | 0.54 | 4.51 | 1.59 | 0.78 | 0.72 |
| SimpleCIL (Zhou et al., 2025a) | 81.06 | 74.48 | 76.84 | 74.48 | 52.06 | 40.13 | 45.11 | 40.13 | 82.13 | 75.58 | 78.62 | 75.58 |
| CoOp (Zhou et al., 2022) | 60.73 | 37.52 | 54.20 | 39.77 | 21.24 | 6.29 | 16.21 | 6.82 | 45.93 | 23.11 | 39.33 | 24.89 |
| ZS-CLIP (Radford et al., 2021) | 83.37 | 77.17 | 79.57 | 77.17 | 38.43 | 26.43 | 31.12 | 26.43 | 79.42 | 72.11 | 74.95 | 72.11 |
| L2P (Wang et al., 2022c) | 75.97 | 66.52 | 72.82 | 66.77 | 51.40 | 39.39 | 48.91 | 42.83 | 82.82 | 74.54 | 79.57 | 73.10 |
| DualPrompt (Wang et al., 2022b) | 76.21 | 66.65 | 73.22 | 67.58 | 52.62 | 40.72 | 49.08 | 42.92 | 82.46 | 74.40 | 79.37 | 73.02 |
| CODA-Prompt (Smith et al., 2023) | 77.69 | 68.95 | 73.71 | 68.05 | 46.49 | 34.13 | 40.57 | 34.13 | 83.34 | 75.71 | 80.38 | 74.17 |
| RAPF (Huang et al., 2024) | 83.56 | 76.63 | 79.61 | 75.92 | 53.78 | 34.97 | 45.37 | 35.74 | 85.23 | 78.21 | 81.91 | 78.62 |
| PROOF (Zhou et al., 2025b) | 83.84 | 78.40 | 81.20 | 78.92 | 56.07 | 43.69 | 48.90 | 43.62 | 83.89 | 77.25 | 80.15 | 76.54 |
| HERMAN | **84.98** | **80.73** | **82.04** | **80.70** | **56.79** | **44.21** | **49.78** | **44.34** | **86.76** | **80.76** | **83.51** | **80.85** |

(a) CIFAR Base0 Inc10     (b) ImageNet-R Base0 Inc10     (c) SUN Base0 Inc10

Figure 2: Incremental performance of different methods. We report the performance gap after the last incremental stage of HERMAN and the runner-up method at the end of the line. All methods utilize the same CLIP pre-trained weights. More figures are shown in the appendix.

## 5.1 IMPLEMENTATION DETAILS

**Dataset.** We follow (Zhou et al., 2025b; 2022; Wang et al., 2022c) to evaluate the performance on nine benchmark datasets that have domain gap to CLIP's pre-training dataset, *i.e.*, CIFAR100 (Krizhevsky, 2009), CUB200 (Wah et al., 2011), ObjectNet (Barbu et al., 2019), ImageNet-R (Hendrycks et al., 2021), FGVCAircraft (Maji et al., 2013), StanfordCars (Krause et al., 2013), Food101 (Bossard et al., 2014), SUN397 (Xiao et al., 2010) and UCF101 (Soomro et al., 2012).

**Dataset Split.** Following (Rebuffi et al., 2017b; Wang et al., 2022c), we use 'B-$m$ Inc-$n$' to split the classes in CIL. $m$ indicates the number of classes in the first stage, and $n$ represents that of every following stage. The dataset split is adapted following Zhou et al. (2025b). We follow (Rebuffi et al., 2017b) to randomly shuffle the class order with random seed 1993 for all experiments.

Table 2: Results when all methods use the same textual descriptors generated by different LLMs.

| Method | Textual Descriptors | Aircraft B0 Inc10 $\bar{\mathcal{A}}$ | $\mathcal{A}_B$ | CIFAR B0 Inc10 $\bar{\mathcal{A}}$ | $\mathcal{A}_B$ | UCF B0 Inc10 $\bar{\mathcal{A}}$ | $\mathcal{A}_B$ | CUB B0 Inc20 $\bar{\mathcal{A}}$ | $\mathcal{A}_B$ |
|---|---|---|---|---|---|---|---|---|---|
| ZS-CLIP (Radford et al., 2021) | GPT-5 Generated | 31.33 | 22.69 | 84.07 | 75.50 | 80.58 | 71.55 | 79.97 | 68.43 |
| RAPF (Huang et al., 2024) | GPT-5 Generated | 55.71 | 32.96 | 88.35 | 81.31 | 94.52 | 86.69 | 82.48 | 65.95 |
| HERMAN | GPT-5 Generated | **66.70** | **58.84** | **89.02** | **83.21** | **97.69** | **95.72** | **84.71** | **78.37** |
| ZS-CLIP (Radford et al., 2021) | QWEN-Plus Generated | 31.25 | 22.54 | 84.24 | 75.68 | 80.63 | 71.55 | 79.89 | 68.31 |
| RAPF (Huang et al., 2024) | QWEN-Plus Generated | 55.75 | 32.92 | 88.41 | 81.32 | 94.49 | 86.60 | 83.13 | 65.02 |
| HERMAN | QWEN-Plus Generated | **66.60** | **58.30** | **89.02** | **83.17** | **94.34** | **90.60** | **84.87** | **79.86** |

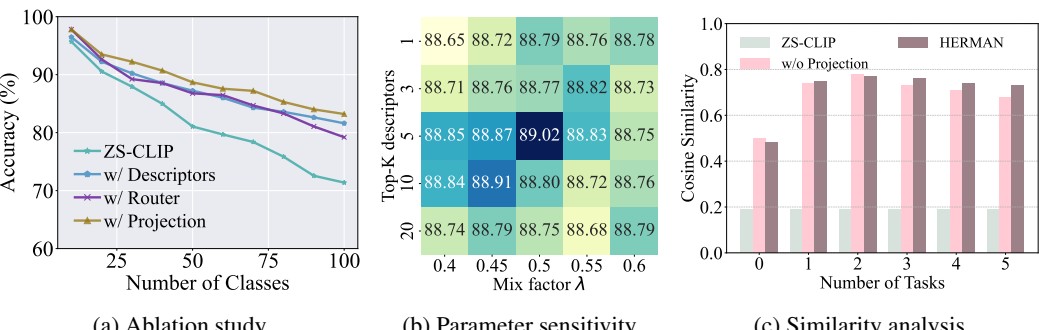

(a) Ablation study      (b) Parameter sensitivity      (c) Similarity analysis

Figure 3: Ablation study, parameter sensitivity and similarity analysis.

**Comparison Methods.** We first compare to SOTA pre-trained model-based CIL algorithms, *e.g.*, L2P (Wang et al., 2022c), DualPrompt (Wang et al., 2022b), CODA-Prompt (Smith et al., 2023) and SimpleCIL (Zhou et al., 2025a). Besides, we also compare to SOTA CLIP-based CIL algorithms, *e.g.*, CoOp (Zhou et al., 2022), RAPF (Huang et al., 2024) and PROOF (Zhou et al., 2025b). As a baseline, we include a variant that fine-tunes CLIP on incremental tasks, denoted as Finetune. All methods are deployed with the same CLIP as initialization.

**Training Details.** The experiments are conducted on NVIDIA 4090 GPUs using PyTorch (Paszke et al., 2019). Following (Huang et al., 2024; Zhou et al., 2025b), all compared methods adopt CLIP with ViT-B/16 pre-trained on LAION-400M (Radford et al., 2021) for *fair comparison*. For vision-only approaches that cannot utilize textual prompts (*e.g.*, L2P, DualPrompt, CODA-Prompt), we initialize them with CLIP's visual branch. In HERMAN, we use an SGD optimizer with a batch size of $64$ for 20 epochs, with the learning rate decaying from $0.05$ according to the schedule. We set Top-$k = 5$, $\lambda = 0.5$ for embedding combination, and $\rho = 0.9, \delta = 0.9$ for the update of the weight of the router. GPT-5 (OpenAI, 2025) is employed to generate textual descriptors.

**Evaluation Metrics.** Following Rebuffi et al. (2017b); Zhou et al. (2025b), we use $\mathcal{A}_t$ to represent the model's accuracy after the $t$-th task. Specifically, we adopt $\mathcal{A}_T$ (the performance after the last task) and $\bar{\mathcal{A}} = \frac{1}{T} \sum_{t=1}^{T} \mathcal{A}_t$ (average performance along incremental tasks) as measurements.

## 5.2 Benchmark Comparison and Further Analysis

**Benchmark Comparison.** We first compare HERMAN with state-of-the-art methods on benchmark datasets, as reported in Tab. 1 and Fig. 2. HERMAN consistently outperforms existing approaches by $1\%$–$5\%$. The Finetune baseline performs the worst, reflecting its severe forgetting of fine-grained class features. Visual prompt-based methods (*e.g.*, L2P, DualPrompt, CODA-Prompt) are limited by their inability to leverage textual semantics, while the textual prompt tuning method CoOp suffers from forgetting of learned prompts, leading to suboptimal results. Notably, HERMAN even surpasses the exemplar-based method PROOF, further demonstrating its effectiveness. This advantage stems from hierarchical representation matching, which aligns multi-level textual descriptors with visual features, enabling stronger cross-modal alignment and more stable knowledge retention.

**Robustness for Textual Descriptors.** We investigate how HERMAN performs using different LLM-generated textual descriptors. Specifically, we apply the same prompt for GPT-5 (OpenAI, 2025) and QWEN-Plus (Yang et al., 2025) to generate textual descriptors, and report the results in Tab. 2. Remarkably, even with the same prompt, HERMAN consistently outperforms the other methods. This demonstrates that HERMAN not only benefits from hierarchical representations but also exhibits robustness in leveraging textual descriptors generated by different LLMs.

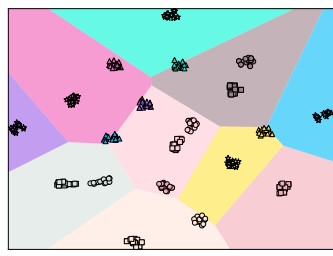 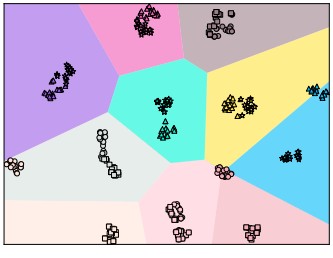 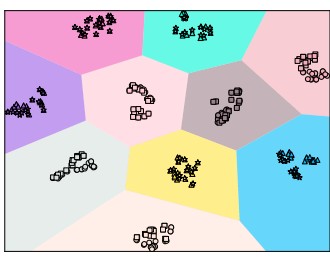

|(a) Matching with 4 layers|(b) Matching with 8 layers|(c) Matching with 12 layers|

Figure 4: We visualize 5 classes of the first task in light colors. In this case, visual and textual embeddings are represented by circles and squares. For the second task, 5 classes are shown in vivid colors, with visual and textual embeddings represented by triangles and stars, respectively. All embeddings are visualized after the completion of all learning tasks.

**Ablation Study.** We conduct ablation experiments on CIFAR100 B0 Inc10 to evaluate the contribution of each component in HERMAN, with results shown in Fig. 3a. **'ZS-CLIP'** serves as the baseline and performs the worst due to distributional shifts. **'w/ Descriptors'** introduces LLM-generated textual descriptors (Eq. 4) and significantly boosts performance, showing the benefit of richer features. **'w/ Router'** adds a router without constraints (Eq. 6) but provides little or no improvement. Finally, **'w/ Projection'**, the full HERMAN model with projection-constrained router updates (Eq. 9), achieves the best results, confirming the effectiveness of each component.

**Parameter Robustness.** We evaluate robustness on CIFAR100 B0 Inc10 by varying two parameters: the number of top-$K$ textual descriptors and the mixing factor $\lambda$ between templated and unified embeddings. Specifically, we set $K \in \{1, 3, 5, 10, 20\}$ and $\lambda \in \{0.40, 0.45, 0.50, 0.55, 0.60\}$. The average performance $\bar{\mathcal{A}}$ is shown in Fig. 3b. Results indicate that too few descriptors fail to capture sufficient hierarchical information, while too many introduce noise. In contrast, $\lambda$ shows stable performance across values. Overall, HERMAN exhibits strong robustness to these hyperparameter choices, and we recommend $K = 5$ and $\lambda = 0.5$ as default settings.

**Update Router without Forgetting.** We conduct experiments on CIFAR100 B50 Inc10 to investigate the role of $\mathbf{P_{old}}$ in maintaining the routing pattern subspace learned from earlier tasks. The cosine similarity between the visual and textual embeddings of the first-task classes is averaged, as shown in Fig. 3c. **'ZS-CLIP'** freezes the model without adaptation, yielding no similarity improvement. **'w/o Projection'**, which updates the router without projecting $\mathbf{P_{old}}$, initially adapts to downstream tasks but quickly deteriorates due to the lack of constraints to retain prior routing patterns. In contrast, **'HERMAN'** leverages the projection mechanism during router updates and sustains the highest cosine similarity as tasks are added incrementally, highlighting the effectiveness of the projection constraint in mitigating catastrophic forgetting.

**Effect of Hierarchical Representations Matching.** We investigate the impact of hierarchical information on cross-modal alignment using CIFAR100 B0 Inc10 with t-SNE (Maaten & Hinton, 2008), as shown in Fig. 4. Alignment is performed with textual descriptors extracted from 4 to 12 intermediate layers. As more layers are incorporated, embeddings of visual and textual modalities cluster more tightly and their gap progressively narrows, indicating stronger alignment. This trend demonstrates that richer hierarchical information consistently enhances cross-modal matching.

## 6 CONCLUSION

We present HERMAN, a hierarchical representation matching method for CLIP-based CIL. By enriching the semantic space with LLM-generated descriptors, aligning them with hierarchical visual representations, and introducing an adaptive routing mechanism with projection-based updates, our method effectively enhances discrimination and mitigates catastrophic forgetting. Extensive experiments across multiple benchmarks validate the robustness and effectiveness of our method.

**Limitations and Future Work.** Our method relies on LLMs to generate textual descriptors for hierarchical representation matching, which may be less effective in scenarios where LLMs fail to generalize. Future directions include leveraging not only the class token but also patch-level tokens to capture richer visual–textual correspondences.

## ETHICS STATEMENT

This work adheres to the ICLR Code of Ethics. Our study is purely methodological and does not involve human subjects, personally identifiable information, or sensitive data. All datasets used are publicly available benchmark datasets, and their usage strictly follows the respective licenses. The proposed method does not raise foreseeable risks of harm, privacy infringement, or discrimination, and all experiments are conducted in compliance with accepted standards of research integrity. We believe our contributions align with the principles of responsible stewardship, fairness, and transparency, and we commit to releasing the source code and results upon acceptance to further support reproducibility and openness.

## REPRODUCIBILITY STATEMENT

We are committed to ensuring the reproducibility of our work. All model architectures, training details, and hyperparameter settings are clearly described in the main text (Sec. 4, Sec. 5) and Appendix (Sec. 6). The datasets used in our experiments are publicly available. To further promote transparency, we will release our source code and all experimental results upon acceptance, enabling the community to fully reproduce and extend our findings.

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

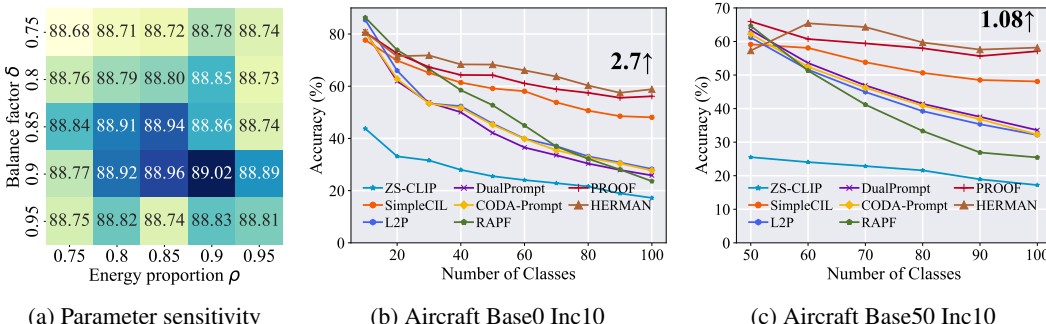

(a) Parameter sensitivity     (b) Aircraft Base0 Inc10     (c) Aircraft Base50 Inc10

Figure 5: Parameter sensitivity analysis and incremental performance of different methods. We report the performance gap on Aircraft after the last incremental stage of HERMAN and the runner-up method at the end of the line. All methods utilize the same CLIP pre-trained weights.

# APPENDIX

## A  USE OF LLMS

In this work, large language models (LLMs), such as ChatGPT, were used solely as general-purpose assistive tools. Specifically, LLMs were employed to help polish the writing style, rephrase sentences for improved clarity, and check grammar consistency. They were not involved in research ideation, the design of the proposed method, the execution of experiments, or the analysis of results. All technical contributions, including problem formulation, methodological design, experimental implementation, and result analysis, were conceived and carried out entirely by the authors.

## B  FURTHER PARAMETER SENSITIVITY ANALYSIS

We further evaluate the robustness of HERMAN with respect to the balance factor $\delta$ in the projection step and the mixing coefficient $\rho$ in Eq. 9 and Eq. 10. Experiments are conducted on CIFAR100 B0 Inc10, with results summarized in Fig. 5a. As shown, varying $\delta$ across $\{0.75, 0.80, 0.85, 0.90, 0.95\}$ and $\rho$ across $\{0.75, 0.80, 0.85, 0.90, 0.95\}$ leads to only minor fluctuations in performance, all remaining within a narrow band around the reported average accuracy. This demonstrates that the model is not overly sensitive to either parameter: $\delta$ effectively controls the proportion of preserved subspace without destabilizing learning, while $\rho$ balances stability and plasticity in a consistent manner. Overall, HERMAN shows strong robustness under different choices of $\delta$ and $\rho$, with default settings $\delta = 0.9$ and $\rho = 0.9$ adopted throughout the main experiments.

## C  EXTENDED BENCHMARK RESULTS

To provide a more comprehensive evaluation, we report incremental performance on nine benchmark datasets in Figs. 5b, 5c and 6 to 13. For each dataset, we explicitly highlight the performance gap between HERMAN and the runner-up method at the final incremental stage using arrows and annotations in the figures. As observed, HERMAN consistently achieves higher accuracy on the last task, outperforming the second-best method by margins ranging from $0.43\%$ to $4.62\%$. These consistent gains across diverse datasets, including fine-grained recognition tasks such as Aircraft, Cars, and CUB, large-scale benchmarks such as ImageNet-R and ObjectNet, and more heterogeneous domains such as SUN and UCF, demonstrate the broad applicability of our approach. The improvements confirm that hierarchical representation matching strengthens cross-modal alignment and enhances robustness against forgetting, enabling the model to retain fine-grained knowledge while flexibly adapting to new categories in continual learning.

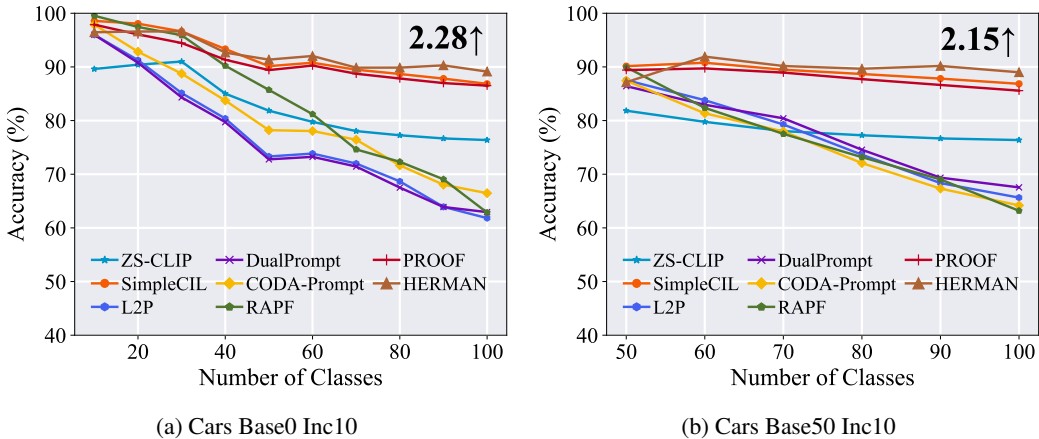

(a) Cars Base0 Inc10

(b) Cars Base50 Inc10

Figure 6: Incremental performance of different methods. We report the performance gap on Cars after the last incremental stage of HERMAN and the runner-up method at the end of the line. All methods utilize the same CLIP pre-trained weights.

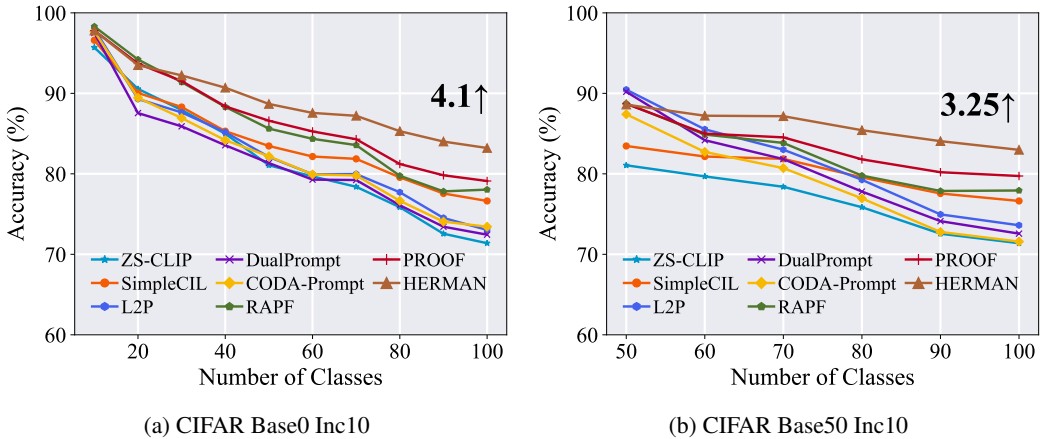

(a) CIFAR Base0 Inc10

(b) CIFAR Base50 Inc10

Figure 7: Incremental performance of different methods. We report the performance gap on CIFAR100 after the last incremental stage of HERMAN and the runner-up method at the end of the line. All methods utilize the same CLIP pre-trained weights.

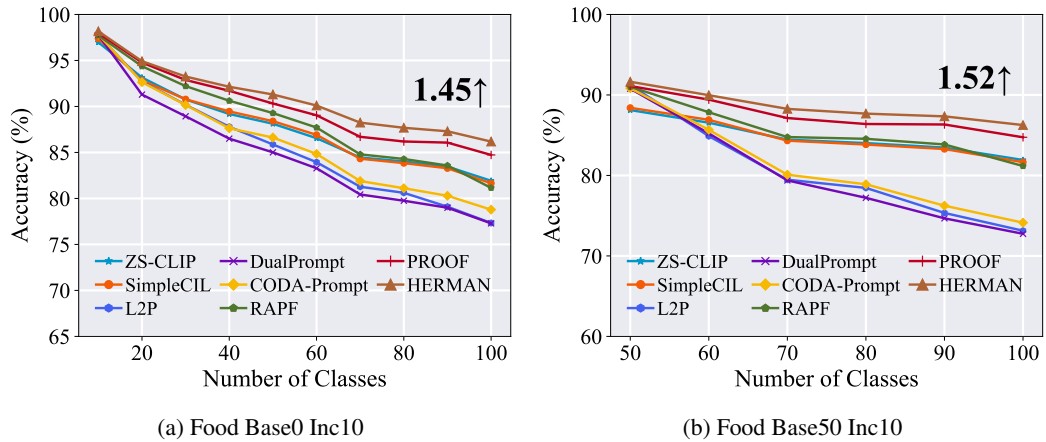

(a) Food Base0 Inc10

(b) Food Base50 Inc10

Figure 8: Incremental performance of different methods. We report the performance gap on Food101 after the last incremental stage of HERMAN and the runner-up method at the end of the line. All methods utilize the same CLIP pre-trained weights.

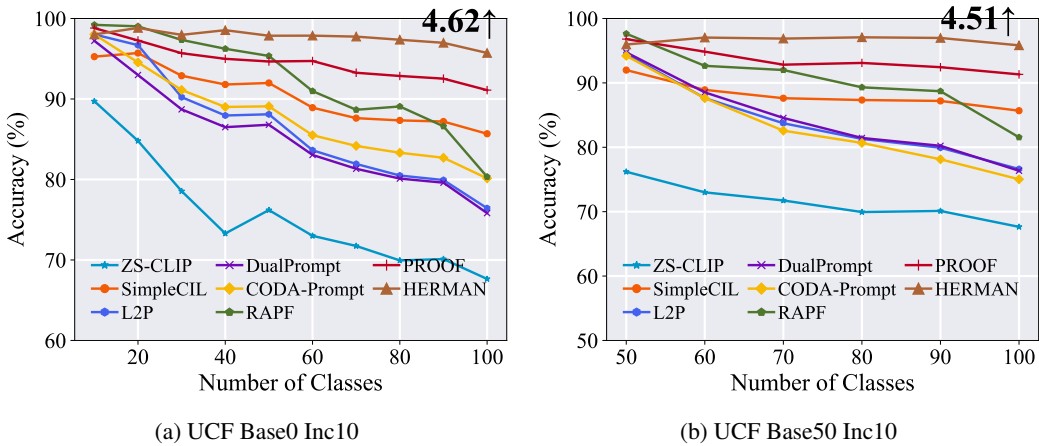

Figure 9: Incremental performance of different methods. We report the performance gap on UCF after the last incremental stage of HERMAN and the runner-up method at the end of the line. All methods utilize the same CLIP pre-trained weights.

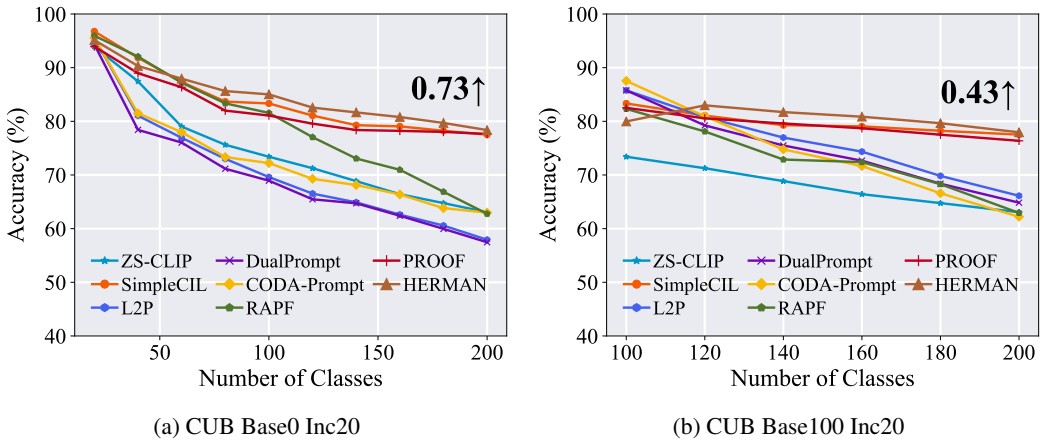

Figure 10: Incremental performance of different methods. We report the performance gap on CUB after the last incremental stage of HERMAN and the runner-up method at the end of the line. All methods utilize the same CLIP pre-trained weights.

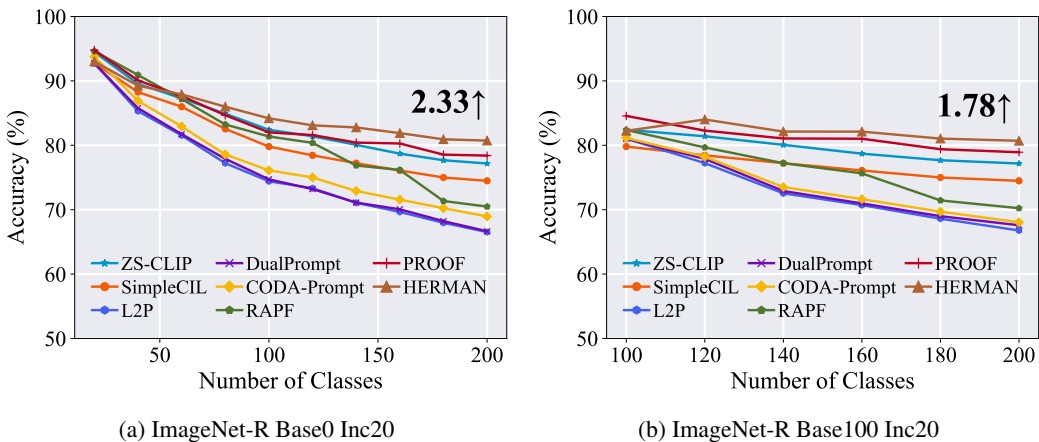

Figure 11: Incremental performance of different methods. We report the performance gap on ImageNet-R after the last incremental stage of HERMAN and the runner-up method at the end of the line. All methods utilize the same CLIP pre-trained weights.

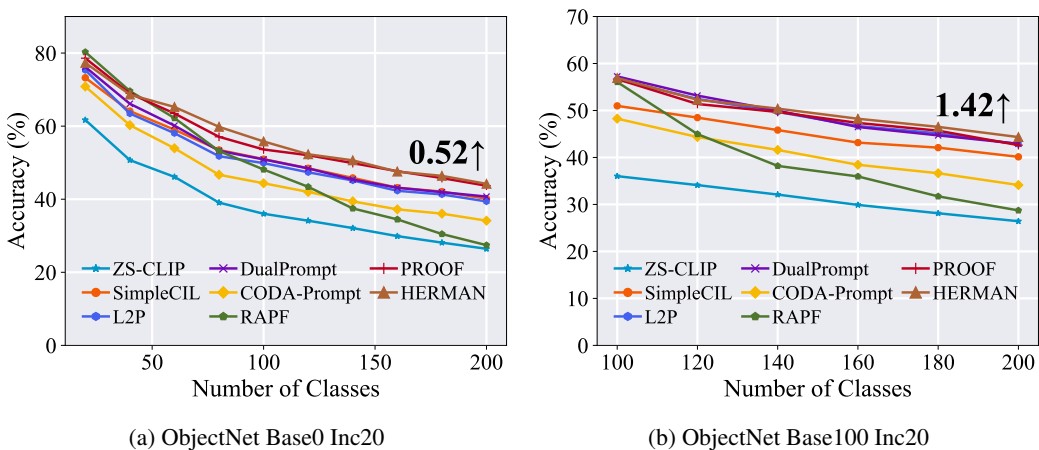

(a) ObjectNet Base0 Inc20          (b) ObjectNet Base100 Inc20

Figure 12: Incremental performance of different methods. We report the performance gap on ObjectNet after the last incremental stage of HERMAN and the runner-up method at the end of the line. All methods utilize the same CLIP pre-trained weights.

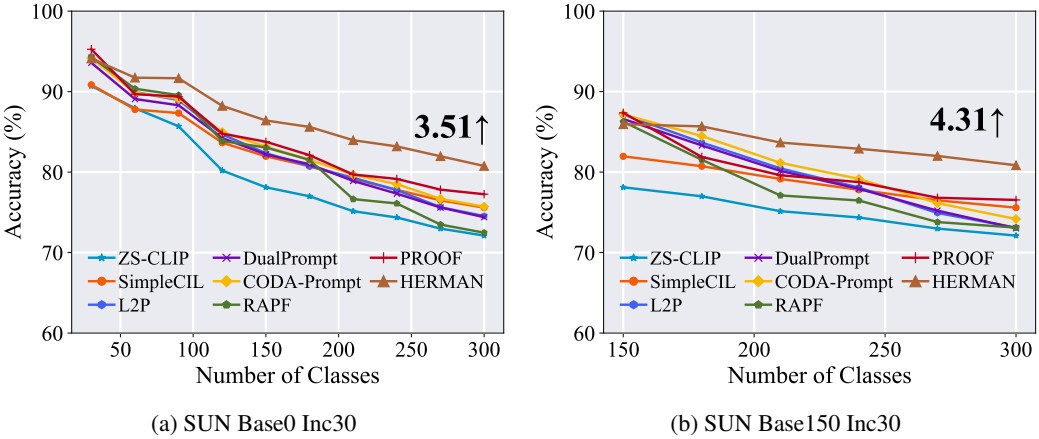

(a) SUN Base0 Inc30          (b) SUN Base150 Inc30

Figure 13: Incremental performance of different methods. We report the performance gap on SUN after the last incremental stage of HERMAN and the runner-up method at the end of the line. All methods utilize the same CLIP pre-trained weights.

