# OpenReview forum: "Hierarchical Representation Matching for CLIP-based Class-Incremental Learning"
_ICLR.cc/2026/Conference — ICLR 2026 Conference Withdrawn Submission_

### Official Review · Reviewer_2m6n · 2025-10-15

**Soundness:** 3
**Presentation:** 3
**Contribution:** 2
**Rating:** 4
**Confidence:** 5

**Summary:**

This paper proposes HERMAN (HiErarchical Representation MAtchiNg) for CLIP-based class-incremental learning (CIL). It (i) uses an LLM to generate hierarchical textual descriptors per class, (ii) aligns these with intermediate CLIP layer features via Top-K matching, (iii) introduces an adaptive router to weight layers. Across nine benchmarks, HERMAN reports consistent gains over prior CIL methods.

**Strengths:**

- The paper is easy to follow.
- Across nine benchmarks, HERMAN reports consistent gains over prior CIL methods.
- Reproducibility details are clear.

**Weaknesses:**

- The paper’s central motivation is to leverage CLIP’s multi-layer representations and LLM-generated, multi-granularity textual descriptors. However, these design choices are largely orthogonal to the core challenges of class-incremental learning (CIL), so the method does not appear to be specifically tailored for CIL. While the authors do address forgetting via the routing update, this seems more like a localized fix than a CIL-driven design principle.
- The novelty of this paper is limited. (i) The utilization of fine-grained description to enhance the representation has been studied, such as [a][b]. (ii) CLIP's multi-layer representations are also studied, such as [c]. The proposed adaptive hierarchy routing is similar to the Trainable Importance Estimator in [c].
- More recent works should be compared, such as [d] [e].
- For the ablation study, (i) more datasets should be applied, and (ii) results for all pairwise component combinations in addition to the full method should be reported.


[a] DEMOCRATIZING FINE-GRAINED VISUAL RECOGNITION WITH LARGE LANGUAGE MODELS, ICLR 2024.
[b] Does VLM Classification Benefit from LLM Description Semantics? AAAI 2025.
[c] LEVERAGING REPRESENTATIONS FROM INTERMEDIATE ENCODER-BLOCKS FOR SYNTHETIC IMAGE DETECTION, ECCV 2024.
[d] Multi-Granularity Class Prototype Topology Distillation for Class-Incremental Source-Free Unsupervised Domain Adaptation, CVPR 2025.
[e] Mind the Gap: Preserving and Compensating for the Modality Gap in CLIP-Based Continual Learning, ICCV 2025.

**Questions:**

See the detailed comments in weaknesses.

---

### Official Review · Reviewer_uUw4 · 2025-10-28

**Soundness:** 2
**Presentation:** 3
**Contribution:** 2
**Rating:** 4
**Confidence:** 4

**Summary:**

This paper presents HERMAN (HiErarchical Representation MAtchiNg), a method for class-incremental learning (CIL) built upon CLIP. The approach leverages hierarchical visual-textual representations by aligning CLIP’s intermediate visual features with LLM-generated textual descriptors of varying granularity (from coarse to fine). To combine these multi-level representations, the authors propose an adaptive routing mechanism with a projection-based update to preserve stability across incremental tasks. Experiments on nine CIL benchmarks show consistent improvements over prior CLIP-based methods such as RAPF and PROOF.

**Strengths:**

- The paper is well-written and easy to follow, with clear motivation and problem formulation.

- The hierarchical descriptor idea is intuitively appealing and effectively illustrated.

- The framework is modular (descriptor generation $\rightarrow$ alignment $\rightarrow$ routing), making it conceptually straightforward to implement.

- The experiments are comprehensive, covering nine benchmarks with reasonable ablations and sensitivity analyses.

- The comparison baselines are mostly fair, as all methods use the same pre-trained CLIP backbone.

**Weaknesses:**

- The core technical contribution lacks novelty. Aligning CLIP’s intermediate layers with hierarchical or multi-level textual cues is similar to prior work that already explored hierarchical or layer-wise multimodal alignment [1, 2].

- The hierarchical textual descriptors depend heavily on LLMs (GPT-5, Qwen-Plus), yet the paper offers no analysis of descriptor quality, variability, or noise sensitivity. The LLM essentially produces descriptive phrases, which may not always be semantically consistent or discriminative.

- The adaptive router with a projection constraint is conceptually close to prior projection/subspace stabilization in continual learning, where updates are projected to preserve subspaces important for past tasks [3, 4].

- There is no discussion of computational overhead or the cost of descriptor generation and multi-layer matching, which are relevant for practical deployment.


[1] Visual Query Tuning: Towards Effective Usage of Intermediate Representations for Parameter and Memory Efficient Transfer Learning. In CVPR 2023.

[2] Multi-Layer Visual Feature Fusion in Multimodal LLMs. In CVPR 2025.

[3] Gradient Projection Memory for Continual Learning. In ICLR 2021.

[4] Efficient Lifelong Learning with A-GEM. In ICLR 2019.

**Questions:**

See Weaknesses.

---

### Official Review · Reviewer_jiuj · 2025-10-30

**Soundness:** 3
**Presentation:** 3
**Contribution:** 2
**Rating:** 4
**Confidence:** 4

**Summary:**

This work proposes Hierarchical Representation Matching for CLIP-based CIL in order to dynamically exploit hierarchical representations. Furthermore, to select the most suitable semantic hierarchy, it introduces an adaptive routing mechanism. Plus, additional techniques are introduced to stabilize the routing process.

**Strengths:**

1. It utilizes LLM to generate multiple text descriptors ( from coarse to fine-grained).
2. Adaptive routing mechanisms are beneficial for adjusting the expression and mutual trade-offs of features at different levels.

**Weaknesses:**

1. Regarding the route update issue, intuitively, using projection constraints can indeed reduce subspace overlap during optimization. However, it seems the ablation experiments don't cover this part of the update. Could the authors demonstrate this ablation experiment? I see Figure 3(c) is about similarity, mentioning this ablation, but it only shows similarity, not accuracy (neither the accuracy of old knowledge nor the overall accuracy).

2. The router could be considered for direct freezing. If this reveals a performance difference, it proves that router optimization is necessary to some extent; otherwise, it doesn't demonstrate this. I hope the authors can provide experiments in this area.

**Questions:**

1. Regarding $\beta$, I'd like to ask if there's a corresponding distribution for beta that can be shown? After all, the distribution of $\beta $ determines whether this is a true multi-layered problem, or whether only one or two layers are activated.

---

### Official Review · Reviewer_dGS6 · 2025-11-02

**Soundness:** 3
**Presentation:** 3
**Contribution:** 2
**Rating:** 4
**Confidence:** 4

**Summary:**

HERMAN enhances CLIP for class-incremental learning via hierarchical textual descriptors from LLMs, aligned with multi-layer visual features, and an adaptive router updated with projections to reduce forgetting. It outperforms SOTA on benchmarks such as CIFAR-100 and ImageNet-R.

**Strengths:**

1.Integrates LLMs naturally for enriching semantics in CIL.
2.The adaptive routing mechanism helps balance coarse- and fine-grained cues.
3.The projection-based router update is a nice touch for stability in CIL.

**Weaknesses:**

1.The novelty is limited, as hierarchical matching builds on existing ideas such as multi-layer features in CLIP (e.g., Enhancing CLIP with GPT-4) and prompt hierarchies, which don't introduce fundamentally new concepts.

2.Similar hierarchical prompting appears in works such as Menon & Vondrick (2023) and Khattak et al. (2023), and the routing is reminiscent of a mixture-of-experts without strong differentiation.

3.The projection for the router feels like a minor tweak on subspace preservation techniques.

4.Experiments lack comparisons to more recent CIL methods beyond PROOF, such as those using advanced replay or distillation in vision-language settings.

5.The method's complexity (LLM generation, routing, projection) may not justify the modest gains over simpler baselines like SimpleCIL.

6.How sensitive is performance to the choice of LLM (e.g., GPT-5 vs. others)? The paper shows robustness, but what about smaller models?

7.Could you clarify the computational overhead of recursive descriptor generation and routing?

8.Why not compare to non-CLIP CIL methods like iCaRL or DER for a broader context?

9.How does HERMAN handle domains far from CLIP's pre-training, beyond the reported benchmarks?

10.What is the runtime cost of LLM descriptor generation per class? Is it feasible for large-scale CIL?

11.Could you provide t-SNE visualizations for additional datasets to demonstrate the alignment benefits?

12.How does HERMAN perform with frozen text encoders or minor CLIP variants?

**Questions:**

Please see the weaknesses.

---

### Note · Authors · 2025-11-12

I have read and agree with the venue's withdrawal policy on behalf of myself and my co-authors.